# Sustainability effects of motor control stabilisation exercises on pain and function in chronic nonspecific low back pain patients: A systematic review with meta-analysis and meta-regression

**Daniel Niederer**[1]*, **Juliane Mueller**[2]

**1** Department of Sports Medicine and Exercise Physiology, Institute of Sports Sciences, Goethe University Frankfurt, Frankfurt am Main, Germany, **2** Department of Computer Science / Therapy Sciences, Professorship for Physiotherapy: Exercise Science and Applied Biomechanics, Trier University of Applied Sciences, Trier, Germany

* niederer@em.uni-frankfurt.de

**Data Availability Statement:** All relevant data are within the paper and its Supporting Information files.

## Abstract

### Study design

Systematic review with meta-analysis and meta-regression.

### Background and objectives

We systematically reviewed and delineated the existing evidence on sustainability effects of motor control exercises on pain intensity and disability in chronic low back pain patients when compared with an inactive or passive control group or with other exercises. Secondary aims were to reveal whether moderating factors like the time after intervention completion, the study quality, and the training characteristics affect the potential sustainability effects.

### Methods

Relevant scientific databases (Medline, Web of Knowledge, Cochrane) were screened. Eligibility criteria for selecting studies: All RCTs und CTs on chronic ($\geq$ 12/13 weeks) nonspecific low back pain, written in English or German and adopting a longitudinal core-specific/stabilizing sensorimotor control exercise intervention with at least one pain intensity and disability outcome assessment at a follow-up (sustainability) timepoint of $\geq$ 4 weeks after exercise intervention completion.

### Results and conclusions

From the 3,415 studies that were initially retrieved, 10 (2 CTs & 8 RCTs) on N = 1081 patients were included in the review and analyses. Low to moderate quality evidence shows a sustainable positive effect of motor control exercise on pain (SMD = -.46, Z = 2.9, p < .001) and disability (SMD = -.44, Z = 2.5, p < .001) in low back pain patients when compared to

**Funding:** The authors received no specific funding for this work.

**Competing interests:** The authors have declared that no competing interests exist.

any control. The subgroups' effects are less conclusive and no clear direction of the sustainability effect at short versus mid versus long-term, of the type of the comparator, or of the dose of the training is given. Low quality studies overestimated the effect of motor control exercises.

## Introduction

A multitude of hypothesized and confirmed risk factors for both the onset and chronification of nonspecific low back pain is available in the literature. Beyond psychological and social factors [1], neuromuscular factors (i.e. deficits or impairments) are particularly named [2,3]. Neuromuscular impairments may be successfully treated. Target-oriented interventions to improve neuromuscular deficits, in particular sensorimotor training, is one of the most established therapy form in low back pain treatment [4,5]. Motor-control exercises [5] and Pilates-based stabilization exercises [6] have been shown to be superior to minimal intervention and provide at least similar outcomes to other forms of exercises [5,6]. Core-stability exercises [7] and back pain-oriented stabilization exercises [8] are more effective than general exercises In general, strength/resistance and coordination/stabilisation exercise programs seem to be superior to other interventions in the treatment of chronic low back pain [4]. Taken together, and proofed in a recent network meta-analysis on the direct comparison of exercise types [9] sensorimotor training is—regarding the outcome pain—one of the most, and—regarding physical function—the most effective active regimens for chronic low back pain treatment. Beyond these short or intermediate-term pre-to-post-intervention effects, motor control exercise is likewise superior to inactivity or minimal intervention in the long-term [5]. Compared to other forms of active exercise, stabilisation and core stability exercise was found to be no more effective than in the long term [7,10].

The various exercises summarized under "sensorimotor/stability/motor control" hinders researchers and practitioners in interpreting conflicting evidence and adopting adequate measures in terms of sensorimotor training. Motor control, sensorimotor, perturbation, neuromuscular, core stability, stabilization, Pilates-based and instability trainings are often used to describe sensorimotor training principles. Musculoskeletal control by afferent sensory, in particular proprioceptive, input, central nervous system integration and optimal motor control to assure functional dynamic joint stability during perturbative situations, are key components of all the training forms described above [11]. Studies using these appropriate muscle recruitment patterns and timing key components as the adequate motor answer on perturbations of a (stable) system as trainings principles may thus be pooled in analyses on motor control stabilisation exercises. Classically, motor control exercises contain a pre-education on deep trunk muscles activation and/or the control of deep muscles activation during exercising. In contrast, different definitions and/or definitions with overlaps to non-dynamic motor control situations are often summarized under the term motor control, the pooled effects of (not only but also) long-term effects may have been over- or underestimated. Furthermore, most of the reviews reported intermediate or long-term effects by aggregating effect sizes with a certain (homogeneous) duration after the randomization. Due to the different intervention durations adopted in the different studies included, long-term effects of the interventions (where the effect are assessed during or immediately after therapy) are thus mixed/pooled with short, intermediate, and long-term sustainability effects (where the effect was assessed after a certain time after the completion of the exercise intervention). It is thus often unclear as to whether 1) reported

long-term effects of motor control stabilisation exercises are based on interventions adopting a rigorous definition of sensorimotor exercises, and 2) if the effects are really based on sustainability effects after intervention completion or rather long-term interventions /where the intervention is implemented until the measurement). Likewise, determining the optimal dose for maximal treatment success (response) is still a matter of debate [12,13].

Against the research deficit highlighted above, the research questions of the present systematic review with meta-analysis and meta-regression are: (1) do motor control stabilisation exercises lead to a sustainable improvement of pain intensity and disability in chronic nonspecific low back pain patients compared to an inactive or passive (no active involvement of the patient, mostly massage therapy, manual therapy, and thermotherapy) control group or compared to other exercises; and (2) to what extent do moderating factors like the duration of the time after the completion of the intervention, the study quality, and the training characteristics affect the potential sustainability effects?

## Methods

### Study design

This secondary data analysis was conducted as a systematic review with meta-analysis and meta-regression. The Preferred Reporting Items for Systematic Reviews and Meta-Analysis (PRISMA [14]) guidelines were followed when conducting and reporting this review.

### Inclusion & exclusion criteria

The inclusion and exclusion criteria were defined with respect to PICO (population, intervention, control/comparator, outcome. The detailed criteria for both the participants and studies are displayed in Table 1.

### Literature research

The literature research was performed using the peer review-based databases PubMed (Medline), Web of Knowledge, and the Cochrane Library. Potentially relevant articles were searched adopting the following Boolean search syntax (example for the PubMed search):

(stabili* OR sensorimotor OR "motor control" OR neuromuscular OR perturbation) AND (exercise OR training OR therapy OR intervention OR treatment) AND ("low back pain" OR lumbalgia OR "lower back pain" OR dorsalgia OR backache OR lumbago OR LBP OR "back pain").

An initial exploratory electronic database search was conducted by two independent reviewers (JM and DN) to define the final search terms. Both reviewers independently conducted the main research afterwards. The herewith identified studies were screened for eligibility using 1) titles and 2) abstracts. The remaining full texts were assessed to ascertain whether they are fulfilling the inclusion and not fulfilling the exclusion criteria. Consensus was used to address any disparities; a third reviewer (N.N.) was asked, if necessary, to address any disparities. After study retrieval, additional studies were identified by manually searching through the reference list (cross-referencing) of the selected articles.

### Data extraction

The included studies were screened for common effect estimators (for pain intensity and disability). Standard mean differences between intervention and comparator effect sizes were calculated based on mean and standard deviation values for the respective scale. Data for the sustainability effects in the short term ($\geq$ 4 weeks $\leq$ 3 months), medium term ($>$ 3 and

**Table 1. Inclusion and exclusion criteria for both the studies and the participants.**

| Criterion | Inclusion | Exclusion |
|---|---|---|
| Study design | Controlled | acute/immediate effects/responses<br>Case studies<br>Case-control, cohort studies<br>Reviews |
| Population | Adults<br>*Non-acute (sub-acute or chronic > 6 weeks of duration at the time of study inclusion)"*<br>non-specific<br>low back pain patients | Children, adolescents <18yrs of age |
| Intervention | motor control<br>core-specific sensorimotor /<br>neuromuscular / sensorimotor / perturbation / core stability<br>stabilization / stabilization exercises/training interventions with a defined completion time | Static (non-dynamic) (motor control) exercises |
| Control/ Comparator | Active or Passive | |
| Outcome | At least one measure of pain (e.g., VAS, NRS, Korff) and/or disability (e.g., ODI, RMDQ, KORFF) | |
| Follow-up length | > 3 weeks after exercise intervention completion | Continued exercise intervention until follow-up meassurement |
| Other | Publication or e-pub before 1st October 2018<br>Language: German & English<br>Full-text availability | |

≤ 12months) and long term (> 12 months) after the exercise intervention completion were collected. All data of interest (descriptive, PICO, interventional details, study quality and risk of bias) were retrieved from the individual study data. For that purpose, a data extraction form, designed for this review, was used. One researcher recorded all the pertinent data from the included articles and the other author independently reviewed the extracted data for its relevance, accuracy and comprehensiveness. Consensus was used to address any disparities; a third reviewer (N.N.) was asked, if necessary, to address any disparities. Authors of studies included in this review who have not reported sufficient details in the published manuscript were personally addressed per e-mail for the provision of further data. Effect estimators (pain intensity and disability) were primarily calculated using the visual analogue scale (VAS) or the numeric rating scale (NRS) or sum score inherent of the scale/assessment tool (0–10 or 0–24 or 0–100), as the calculation of the standard mean differences is scale-independent. For such data, only the direction (lower values mean less pain, less disability) was normalized. For scale-dependent calculations (inverse weighting), z-transformed (0–10) variables were used. Missing standard deviations for the differences were imputed according to the procedure described in Follmann et al. [14].

## Study quality assessment

The methodological quality of all controlled trials included was assessed using the PEDro scale (11 criteria). The PEDro scale is a valid and reliable tool to assess the methodological quality of controlled studies [15]. Each criterion was rated as 1 (definitely yes) or 0 (unclear or no); potential disagreements were discussed between the two authors and then resolved.

## Risk of bias within studies/outcomes

The two review authors (JM and DN) independently rated the risk of bias of the included studies, using the Cochrane Collaboration's tool [16]. Following the Cochrane recommendations,

bias was rated outcome specific and not study specific (Cochrane Handbook Version 5.1.0, Chapter 8.7). The outcomes were graded for risk of bias in each of the following domains: sequence generation, allocation concealment, blinding (participants, personnel, and outcome assessment), incomplete outcome data, selective outcome reporting, and other sources of bias. Each item was rated as "high risk", "low risk", or "unclear risk" of bias. Again, any disagreements were discussed between the raters. If a decision could not be reached after discussion, a third reviewer (N.N.) was included to resolve any conflicts. If applicable. The outcomes' bias were reported pooled for studies.

## Measures of treatment effects—Main effects

The Review Manager 5.3 (RevMan, Version 5.3, Copenhagen: The Nordic Cochrane Centre, The Cochrane Collaboration, 2014) was used for data analyses of the main effects. Standardised means differences and sample sizes were used for data pooling. A random-effects meta-analysis model for continuous outcomes was chosen. For variance description, 95% confidence intervals were calculated; data were displayed using Forrest-plots. To test for overall effects, Z-statistics at a 5% alpha-error-probability level were calculated for: 1. Overall (main) effects and 2. Quantitative subgroup analyses. For the overall effect calculation, each intervention group effects was calculated in contrast to the comparator/control group. In studies with more than two MCE arms, more than one effect estimator contributes to the main calculation. If more than one sustainability timepoint was assessed, the mid-term sustainability effect was selected for the main analysis. For the quantitative subgroup calculations, analyses were performed separately for 2a. sensitivity of time (short-term, mid-term, and long-term sustainability), and 2b. sensitivity of comparator (inactive or passive vs. motor control stabilisation exercises (MCE) and other exercises vs. MCE). For variance description of the subgroup analyses, 90% confidence intervals were calculated; data were displayed using Forrest-plots. To test for overall effects, Z-statistics at a 5% alpha-error level were calculated.

## Measures of treatment effects—Assessment of heterogeneity

Clinical heterogeneity between the study results in effect measures was assessed using $I^2$-statistic. An I-squared value greater than 50% is indicative for substantial heterogeneity [16].

## Measures of treatment effects—Sensitivity meta-regression for dose-response analyses

To counteract the considerable heterogeneity, sensitivity meta-regressions for dose-response analyses and the impact of study quality and risk of bias were conducted. A syntax for SPSS (IBM SPSS 23; IBM, USA) was used (David B. Wilson; Meta-Analysis Modified Weighted Multiple Regression; MATRIX procedure Version 2005.05.23). Inverse variance weighted regression models with random intercepts (random effect model, fixed slopes model) with the dependent variables pain and disability effects (simple pre-post Cohen's ds) and the independent variables intervention duration [weeks], intervention frequency [number of trainings/ weeks], intervention [ratio of the sustainability time / training time], intervention total dose [minutes], and study quality PEDro sum score [points]. Homogeneity analysis (Q and p-values) and meta-regression partial coefficients B (95% confidence intervals and p-values) were calculated.

### Risk of bias across studies

The calculation of the risk of publication bias across all studies is indicated by using funnel plots/graphs.

### Effect estimators' level of evidence

Quality of evidence revealed by the main and subgroup meta-analyses were graded using the tool established by the GRADE working group [17]. Quality evidence was categorized as "very low", "low" "moderate", or "high" (plus interim values).

## Results

### Study selection

The database search was completed in 10/2018. Fig 1 displays the research procedure and the flow of the study selection and inclusion.

### Study characteristics and individual studies' results

Ten (10) studies were included in the qualitative and quantitative sustainability analyses. Their characteristics and main results are displayed in Table 2. For each of the studies included, methodological aspects, participants' characteristics, and key results are displayed. Overall, 1,081 participants with nonspecific chronic low back pain were included.

Two of the studies are controlled trials (CT) [18][19], while the other eight adopted a randomized controlled design (RCT) [20,21,22,23,24,25,26,27]. Main inclusion criterion was (sub-acute chronic) nonspecific low back pain ≥ 6 weeks (1x, [25]), ≥ 8 weeks (1x, [26]), ≥12 weeks (3 x, [20,21,22]), 24 weeks (1x, [18]). The baseline pain (VAS, 0–10 points) ranged from 2.9±0.8 [20] to 6.5±2.1[22]. The effect sizes (Cohens d, MCE only) for the sustainability measures ranged from .27 [19] to 2.6 [27] (pain intensity) and, for disability, from .17 [20] to 1.9 [25]

### Study quality and risk of bias within studies (outcomes)

Both the study quality and risk of bias ratings are displayed in Table 3. Overall study quality was 5/11 to 9/11 points, with a mean of 6.8. As the outcomes were assessed using self-reported questions within the same questionnaires, the risk of bias was reported accumulated per study and not per outcome.

### Main effect estimates

The main effect size estimates of the overall sustainability (4 to 44 weeks after exercise intervention completion) effects of motor control stabilisation exercise in comparison to inactive control, passive treatment or other exercises for the outcomes pain intensity and disability are displayed in Fig 2.

Low to moderate quality evidence indicates that MCE has a larger overall sustainability effect on pain intensity and disability than a passive, inactive or other exercise comparator.

### Grouped effect estimates

Figs 3 to 8 show the main effect estimates results as pooled forest plots, separated for sustainability duration after exercise intervention completion (short-term: Figs 3 and 4, mid-term: Figs 5 and 6, and long-term: Figs 7 and 8), for the type comparator (passive or inactive control,

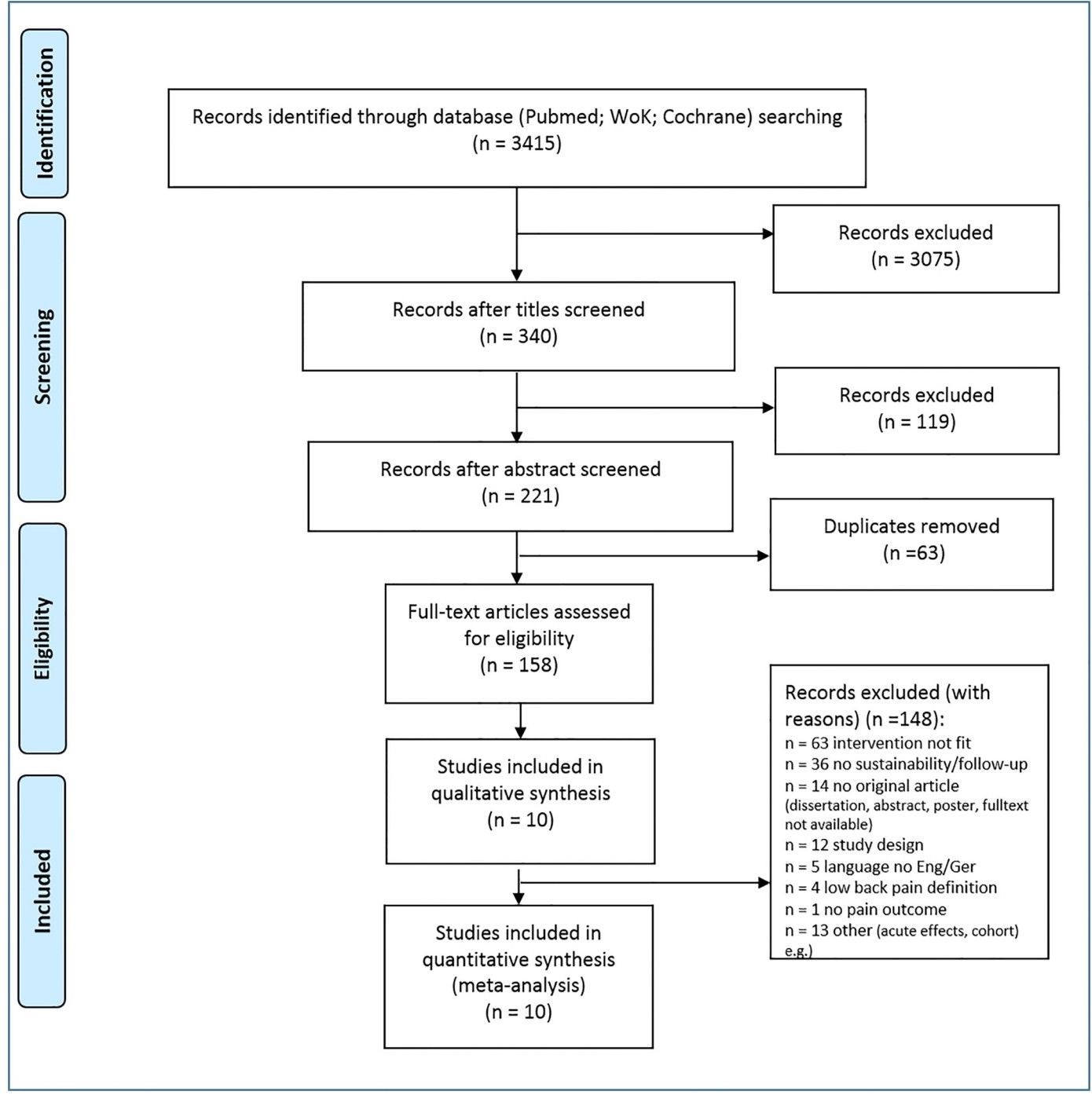

**Fig 1. Research, selection and synthesis of included studies.** n, number; Eng, English, Ger, German; WoK, Web of Knowledge.

Figs 3, 4 and 5; other exercise, Figs 4, 6 and 8), and for the outcomes pain intensity (Figs 3–8, parts—A-) versus disability (Figs 3–8, parts—B-).

Low quality evidence indicates that MCE has no larger short-term sustainability effect on pain intensity than a passive or inactive comparator. Low quality evidence indicates that MCE has a larger short-term sustainability effect on disability than a passive or inactive comparator.

**Table 2. Study characteristics (left columns) and individual studies' results (right columns).** For each of the studies included, methodological aspects, participants' characteristics, and key results are displayed. RCT, randomized controlled trial, CT, controlled trial; MCE, motor control stabilisation exercise, Ctrl, control or comparison group; CLBP, chronic low back pain; N, number; f, female; m, male; SD, standard deviation; Mx, measurement visit number, VAS, visual analogue scale; NRS, numeric rating scale; ODI, Owestry disability index, RMDQ, Roland Morris disability questionnaire.

| First Author, year | Citation | Design, Arms | Main inclusion criterion lbp | N (Total, MCE, C, C2,...) | Age (Mean± SD) | Sex (f/m) | Baseline-pain (Scale, MW, SD if not stated otherwise) | Measurement time points total (N: weeks (if not, stated otherwise) after Baseline) | Primary outcome pain, name, Cohens d, (M0-M1, M0-M2,...) | Primary outcome disability name, Cohens d, (M0-M1, M0-M2,...) |
|---|---|---|---|---|---|---|---|---|---|---|
| Bae, 2018 | [20] | RCT, 2 MCE Ctrl | CLBP ≥ 12 weeks | 36 18 18 | years 32.7±6.1 32.4±11 | 18/20 | VAS (0–10) 2.9±0.8 3.0±1.3 | 4: 4 8 16 | VAS (0–10)1 1.25 1.75 | ODI .19 .17 .24 |
| Critchley, 2007 | [21] | RCT, 3 MCE Ctrl 1 Ctrl 2 | CLBP ≥ 12weeks | 212 72 71 69 | years 44±13 45±12 44±12 | 133/89 | NRS (0–100), mean, 95%CI 67, 61–73 60, 54–66 59, 52–65 | 4: 6 months 12 months 18 months | NRS (0–10) .7 .6 .9 | RMDQ 1.0 .7 .8 |
| Ferreira 2007 | [22] | RCT, 3 MCE Ctrl 1 Ctrl 2 | CLBP ≥ 12 weeks | 240 80 80 80 | years 51.9±15.3 54.8±15.3 54.0±14.4 | 165/75 | VAS (0–10) 6.3±2.0 6.5±2.1 6.2±2.0 | 4: 8 24 48 | VAS (0–10) .9 1 .7 | RMDQ 1.2 1.1 .98 |
| Giesche 2017 | [23] | CT, 2 MCE Ctrl | CLBP ≥24 weeks | 48 25 23 | years 56.5±11.3 60.1±12.2 | 31/17 | NRS (0–10) 4.6±2.0 4.9±2.0 | 4: 2 3 8 | NRS (0–10) .1 .55 .65 | ODI .3 .34 |
| Kofotolis, 2016 | [27] | RCT, 3 MCE Ctrl 1 Ctrl 2 | CLBP ≥12 weeks | 101 37 36 28 | years 41.2±8.5 42.7±6.1 39.1±8.7 | 101/0 | SF-36 pain 38.51±12.62 36.93±15.5 39.4±14.5 | 5: 4 8 12 20 | SF-36 pain (0–100) 1.9 3,2 2,9 | RMDQ .75 1.2 1.1 |
| Macedo, 2012 | [27] | RCT, 2 MCE Ctrl | CLBP ≥ 12 weeks | 158 76 82 | years 48.7±13.7 49.6±16.3 | 102/56 | NRS (0–10) 6.1±2.1 6.1±1.9 | 4: 8 6 months 12 months | NRS (0–10) .95 .95 1.1 | RMDQ .8 .7 .8 |
| Marshall, 2013 | [24] | RCT,2 MCE Ctrl | Recurrent LBP ≥ 12 weeks | 64 32 32 | years 36.2 ± 8.2 36.2 ± 6.2 | 40/24 | VAS (0–10) 3.6 ± 2.1 4.5 ± 2.5 | 3: 8 6 months | VAS (0–10) .9 .76 | ODI .93 .93 |
| Rasmussen-Barr, 2003 | [25] | RCT, 2 MCE Ctrl | LBP sub-acute, chronic or recurrent ≥ 6 weeks | 42 22 20 | years 39± 12 37± 10 | 12, 35 | VAS (0–100), median 25th /75th 33 (27/49) 32 (21/49) | 4: 6 3 months 12 months | VAS (0–100) 1.6 1.3 | ODI 1.9 1.8 |
| Rasmussen-Barr, Eva, 2009 | [26] | RCT, 2 MCE Ctrl | LBP ≥ 8 weeks | 71 36 35 | years 37± 10 40± 12 | 35, 36 | VAS (0–100), VAS (0–100), median 25th /75th 32 (18/75) 38 (23/62) | 5: 8 6 months 12 months 36 months | VAS (0–100), .8 .9 | OSD .9 1.4 |
| Unsgaard-Tondel, 2010 | [19] | CT, 3 MCE Ctrl 1 Ctrl 2 | CLPB | 109 36 36 37 | years 41± 12 43± 10 36± 10 | 33, 76 | NRS (0–10) 3.3 ± 1.3 3.6 ± 1.7 3.3 ± 1.9 | 3: 8 1 year | NRS (0–10) .37 .27 | ODI .9 N.A. |

Low quality evidence indicates that MCE has no larger short-term sustainability effect on pain intensity and disability than other exercises.

Low (to moderate) quality evidence indicates that MCE has no larger mid-term sustainability effect on pain intensity than a passive or inactive comparator or othzer forms of exercise.

**Table 3. Study quality and risk of bias.** PEDro-scale-items: 1) eligibility criteria were specified, 2) participants were randomly allocated to groups, 3) allocation was concealed, 4) the groups were similar at baseline regarding the most important prognostic indicators, 5) there was blinding of all participants, 6) there was blinding of all therapists who administered the therapy, 7) there was blinding of all assessors who measured at least one key outcome, 8) measures of at least one key outcome were obtained from more than 85% of the participants initially allocated to groups, 9) all participants for whom outcome measures were available received the treatment or control condition as allocated or, where this was not the case, data for at least one key outcome was analysed by "intention to treat", 10) the results of between-group statistical comparisons are reported for at least one key outcome, 11) the study provides both point measures and measures of variability for at least one key outcome.

| Number / Item | PEDro | | | | | | | | | | | Sum PEDro | Random sequence generation | Allocation concealment | Performance bias | Detection bias | Attrition bias | Reporting bias | Other bias |
|---|---|---|---|---|---|---|---|---|---|---|---|---|---|---|---|---|---|---|---|
| | 1 | 2 | 3 | 4 | 5 | 6 | 7 | 8 | 9 | 10 | 11 | | | | | | | | |
| Bae, 2018 | 1 | 1 | 0 | 1 | 0 | 0 | 0 | 1 | 1 | 1 | 1 | 6 | unknown | low | high | high | low | low | low |
| Critchley, 2007 | 1 | 1 | 1 | 1 | 0 | 0 | 1 | 0 | 1 | 1 | 1 | 7 | low | low | high | low | high | low | unknown |
| Ferreira 2007 | 1 | 1 | 1 | 1 | 0 | 1 | 1 | 1 | 1 | 1 | 1 | 9 | low | low | high | low | low | low | low |
| Giesche 2017 | 1 | 0 | 0 | 1 | 1 | 0 | 0 | 0 | 1 | 1 | 1 | 5 | high | high | high | high | high | unknown | low |
| Kofotolis, 2016 | 1 | 1 | 0 | 1 | 0 | 0 | 0 | 0 | 1 | 1 | 1 | 5 | unknown | low | high | high | high | low | low |
| Macedo, 2012 | 1 | 1 | 1 | 1 | 0 | 0 | 1 | 1 | 1 | 1 | 1 | 8 | low | low | high | low | low | low | low |
| Marshall, 2013 | 1 | 1 | 1 | 1 | 1 | 0 | 1 | 1 | 1 | 1 | 1 | 9 | unknown | low | high | low | low | low | unknown |
| Rasmussen-Barr, 2003 | 1 | 1 | 0 | 1 | 0 | 0 | 0 | 1 | 0 | 1 | 1 | 5 | unknown | low | high | high | unknown | low | unknown |
| Rasmussen-Barr, 2009 | 1 | 1 | 1 | 1 | 0 | 0 | 0 | 1 | 1 | 1 | 1 | 7 | low | low | high | high | low | low | low |
| Unsgaard-Tondel, 2010 | 1 | 1 | 1 | 1 | 0 | 0 | 0 | 1 | 1 | 1 | 1 | 7 | low | low | high | high | low | low | low |

(Low to) moderate quality evidence indicates that MCE has no larger mid-term sustainability effect on disability than a passive or inactive comparator or than other exercises.

Moderate quality evidence indicates that MCE has no larger long-term sustainability effect on pain intensity than a passive or inactive comparator. (Low to) moderate quality evidence indicates that MCE has no larger long-term sustainability effect on disability than a passive or inactive comparator or than other forms of exercise. Low to moderate quality evidence indicates that MCE has a larger long-term sustainability effect on pain intensity than other exercises.

## Individual studies: Training characteristics

Table 4 summarizes the individual studies' training characteristics. All interventions and the respective comparators are described. The motor control stabilisation exercises are named MCE: [22–24], core stability exercises: [20], stabilization: [21,25,26,28] sensorimotor [18], sling training [19], and Pilates-based exercise [27]. Six out of the ten studies adopted an eight-week intervention, and the mean training time was 53 minutes. Training frequency ranged from 1 [19] to 12 [22] times per week.

## Sensitivity meta-regressions on training characteristics

The results of the five meta-regressions as sensitivity analyses are highlighted in Table 5. The training duration, frequency, total trainings dose and training-to-sustainability ratio showed no impact on the effect size of the primary outcome pain.

The PEDro sum score was negatively associated with the effect size, a study with a score-decrease of 1 point shows an increase in the effect size of .24. Fig 9 illustrates this association.

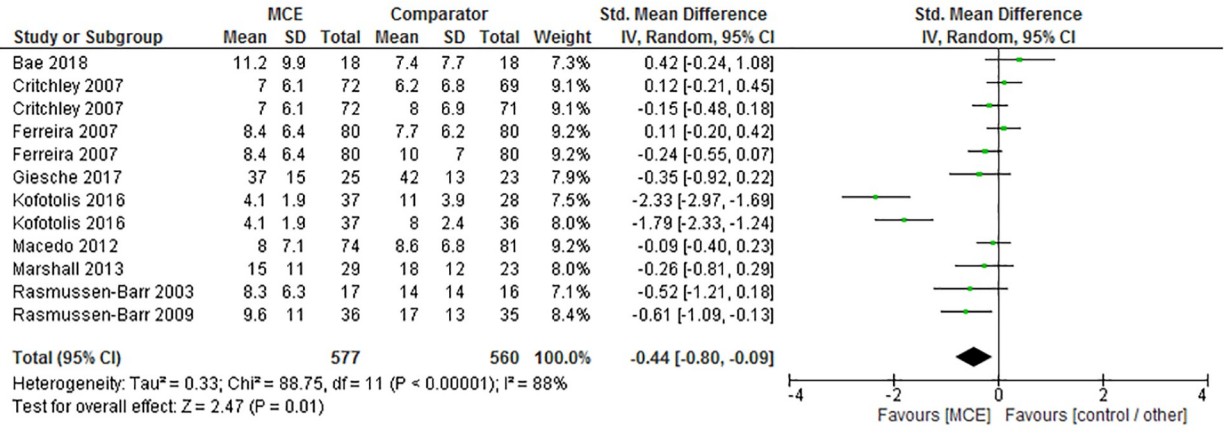

**Fig 2. Pooled main effect size estimates (standardized mean differences) for the outcomes pain intensity (-A-) and disability (-B-).** Overall sustainability effects of motor control stabilisation exercise in comparison to inactive control, passive treatment or other exercises. MCE, motor control stabilisation exercise, SD, standard deviation; CI, confidence interval.

### Risk of bias across studies

The risk of bias across studies (publication bias) is, by means of a funnel plot, highlighted in Fig 10. It reveals an unclear but rather low risk of publication bias.

## Discussion

### Summary of the evidence

We found that motor control stabilisation exercises lead, with low to moderate quality evidence, to a sustainable improvement in pain intensity and disability in chronic non-specific

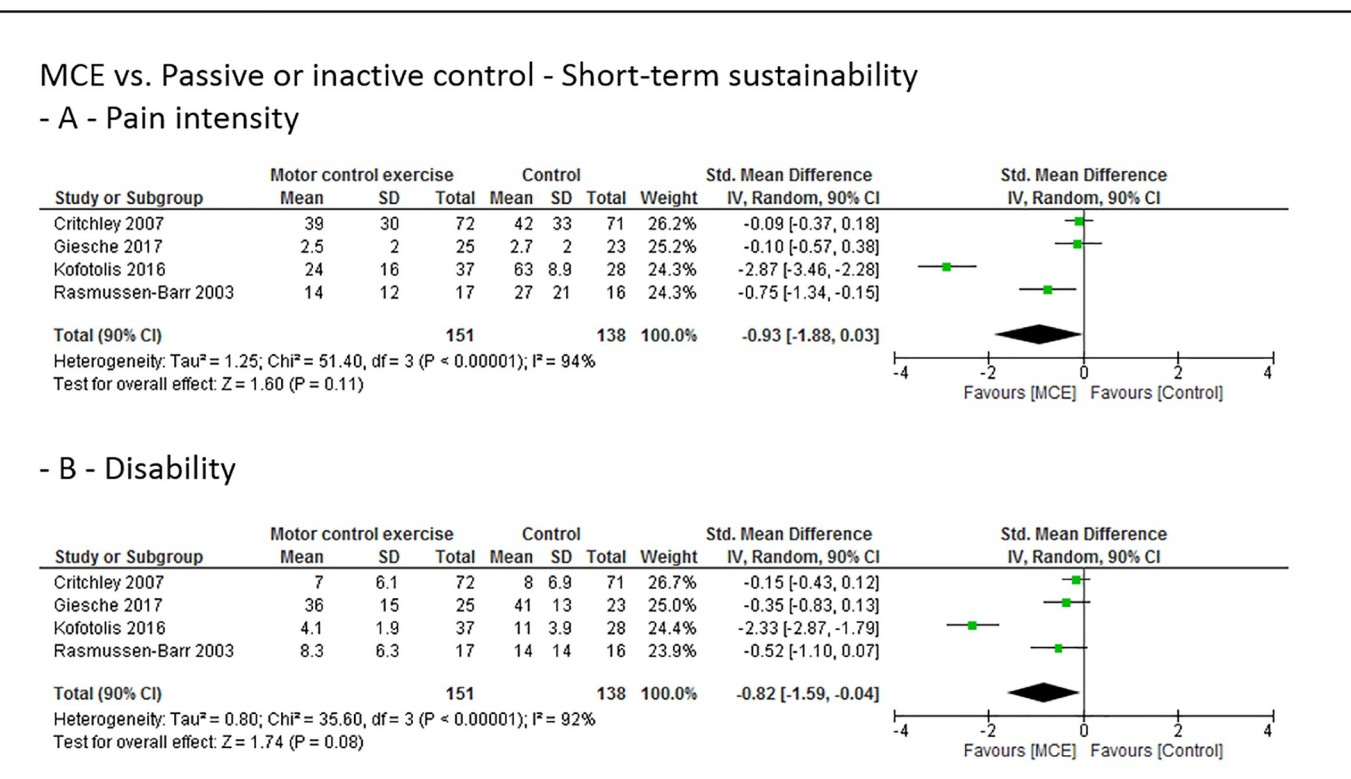

**Fig 3. Pooled effect sizes (standardized mean differences) for the outcomes pain intensity (-A-) and disability (-B-).** Analysis for the short-term sustainability effects of motor control stabilisation exercise in comparison to passive or inactive control. MCE, motor control stabilisation exercise; SD, standard deviation; CI, confidence interval.

low back pain patients compared to an inactive or passive control group or compared to other exercises. Subgroup sensitivity analyses revealed less clear findings: some of the pooled effects reached significance, some not.

The subsequent meta-regression demonstrated that the training duration, frequency, total trainings dose and training-to-sustainability ratio has no impact on the effect size of the primary outcome pain. The PEDro sum score was negatively associated with the effect size and studies with lower quality may overestimate the (sustainability) effects of MCE on pain intensity and disability reduction.

Small overall effects for a larger effect of MCE than other controls/exercises are seen; the subgroup analyses revealed inconsistent results. Here, MCE is at least equivalent to other forms of exercise.

## Comparison with other evidence

To compare our findings with other published evidence, the limitations of MCE training definition and the mix of long-term and sustainability effects highlighted in the introduction must be considered. First, most available evidence focusses on long-term effects (in duration after the randomization) and not on sustainability. Thus, a mix of sustainability effects and effects directly assessed during the intervention or directly after the exercise intervention completion are mixed. Second, not all evidence-based analyses used key components of appropriate

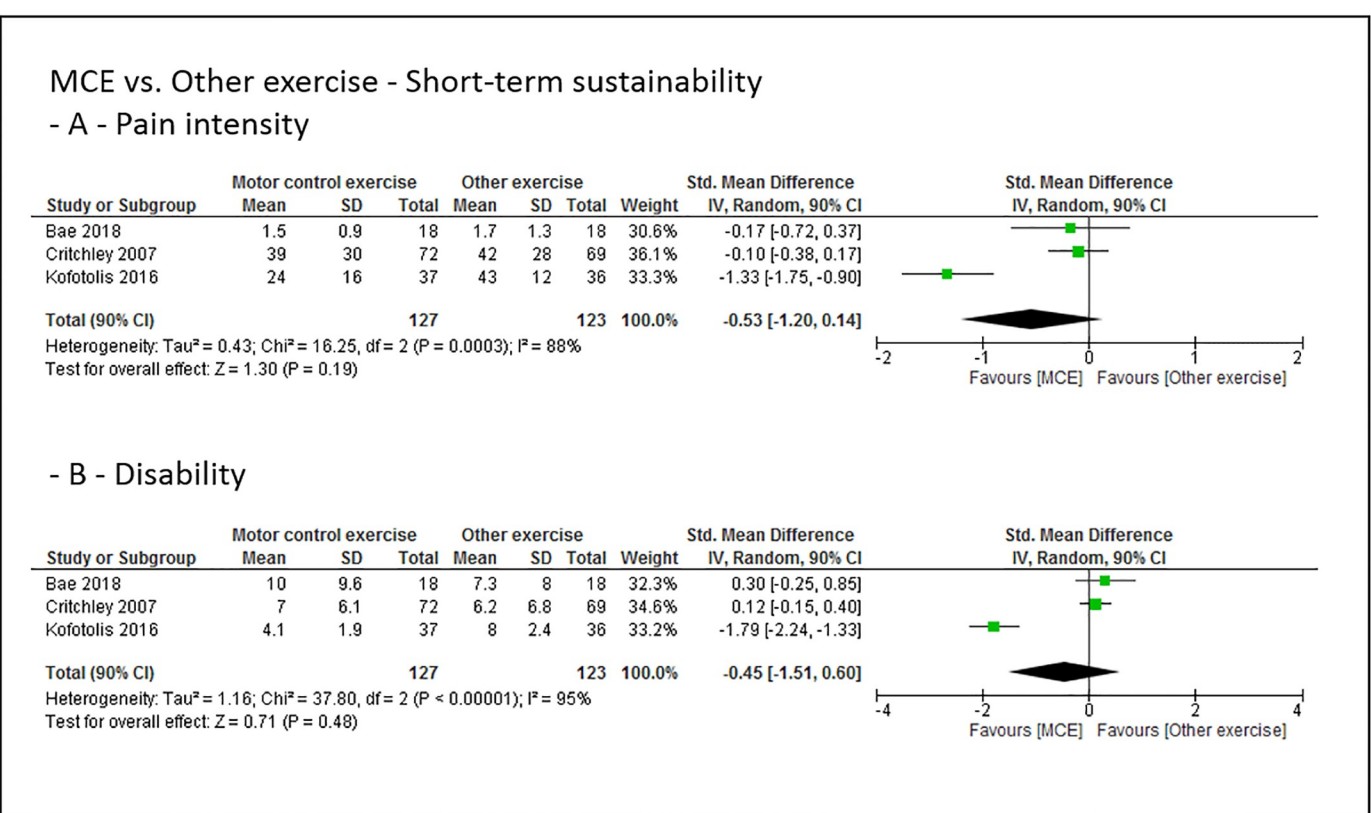

**Fig 4. Pooled effect sizes (standardized mean differences) for the outcomes pain intensity (-A-) and disability (-B-).** Analysis for the short-term sustainability effects of motor control stabilisation exercise in comparison to other exercises. MCE, motor control stabilisation exercise; SD, standard deviation; CI, confidence interval.

muscle recruitment patterns and timing as the adequate motor answer to perturbations of a (stable) system as inclusion criterion for the trainings.

A recent meta-analysis on core-stability trainings in low back pain patients found no follow-up differences in pain reduction between core stability exercise and general exercise [7]. The findings are based on a limited number of studies. A comparable amount of analyses on numerous RCTs adopting stabilization training demonstrated heterogeneous results which are comparable to ours [10]. The authors found a systematic benefit of stabilization exercises on pain intensity when compared with any alternative treatment or control at an intermediate follow-up of 3–12 months and at a long-term follow-up of >12 months. In contrast, they found strong evidence that stabilization exercises are not more effective than any other form of active exercise in the long-term. In the meta-analysis on MCE 5[5], the authors concluded that there is high-quality evidence for no clinically important standardized difference of MCE for pain intensity (when compared to other exercises) or disability (when compared to minimal intervention) at intermediate and long-term follow-ups. When compared to minimal intervention, MCE was found to be in favour of a clinically important effect of pain intensity changes at medium and long-term follow-ups[5].

## Practical relevance

Overall, MCE seems to be slightly more sustainable or at least equivalent to other exercises and slightly more sustainable than passive or inactive treatments in terms of pain intensity

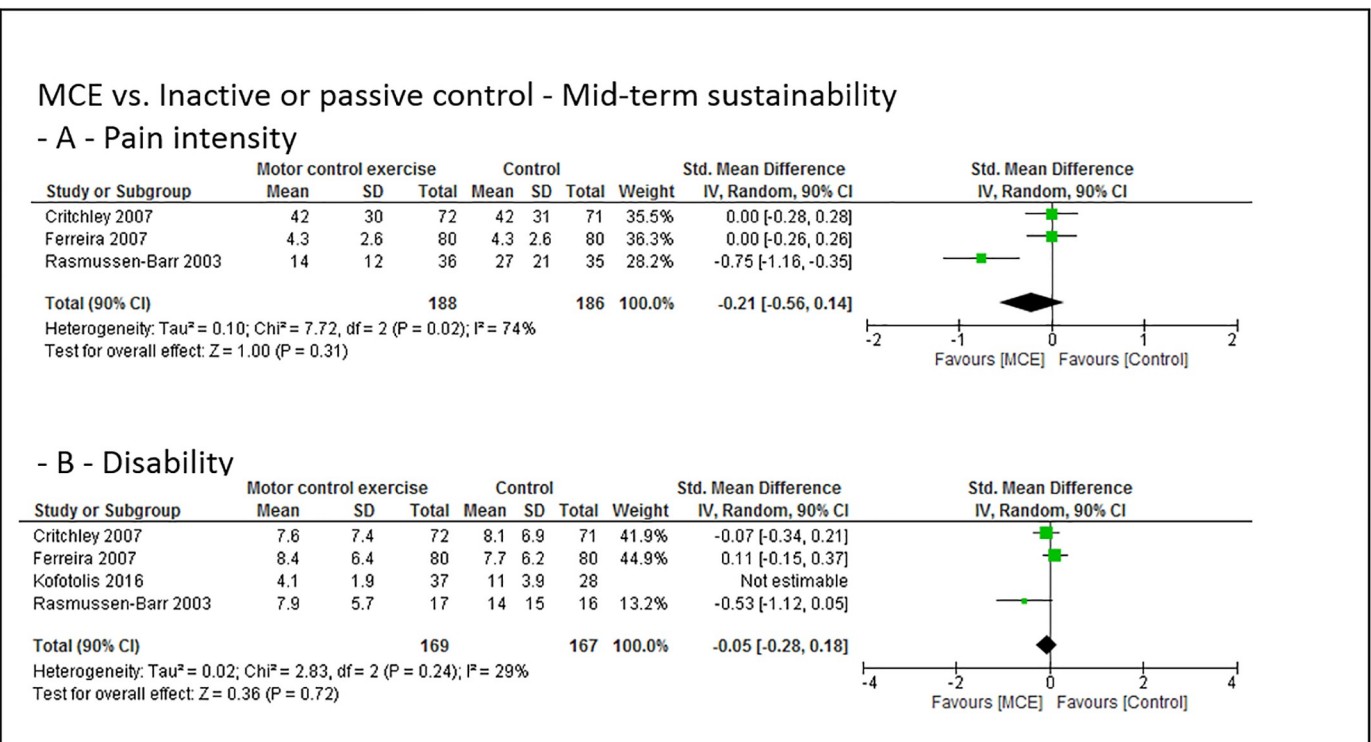

**Fig 5. Pooled effect sizes (standardized mean differences) for the outcomes pain intensity (-A-) and disability (-B-).** Analysis for the mid-term sustainability effects of motor control stabilisation exercise in comparison to passive or inactive control. MCE, motor control stabilisation exercise; SD, standard deviation; CI, confidence interval.

and disability reduction. Although, derived from the quality of evidence of the findings, no grade A recommendation can be provided, but MCE seems to be both effective and safe in the treatment of low back pain. Further, none of the other types of exercise was elicited to be more effective. Therapy should, of course, always be patient-centred and focussed on the individual context and preferences of the patient [29]. Based on the individual patient's preferences, the findings of our review, and proper dose-response relations plus training characteristics, the effects of MCE interventions will most likely be increased in the future.

A suggested underlying mechanism for the general exercise effect in low back pain is mostly seen in the analgesic effect of exercise. Exercise releases beta-endorphins, both spinal and supraspinal, by activating μ-opioid receptors [30]. Following that, an acute sensible decrease in pain is felt. In the long term, exercise and, in particular, sensorimotor motor control training may increase the functional capacity of all involved tissues, leading to a protection against neuromuscular-deficient motor patterns[31].

## Limitations at study and outcome level

A common limitation in exercise trials is the limited possibility to blind the participants. This limitation is increased by the subjective assessment of pain and pain–related function. We showed that a lower study quality is associated with larger effect sizes (MCE groups only). The (overall) effect of the MCE may thus be overestimated. This finding is most likely attributed to the lack of adopting a randomized design (2 studies) as well as to the lack of participant and study personnel blinding or to the fact that most of our significant findings were attributed to only two studies with large effects[26,27].The finding of an overestimation of the effect in

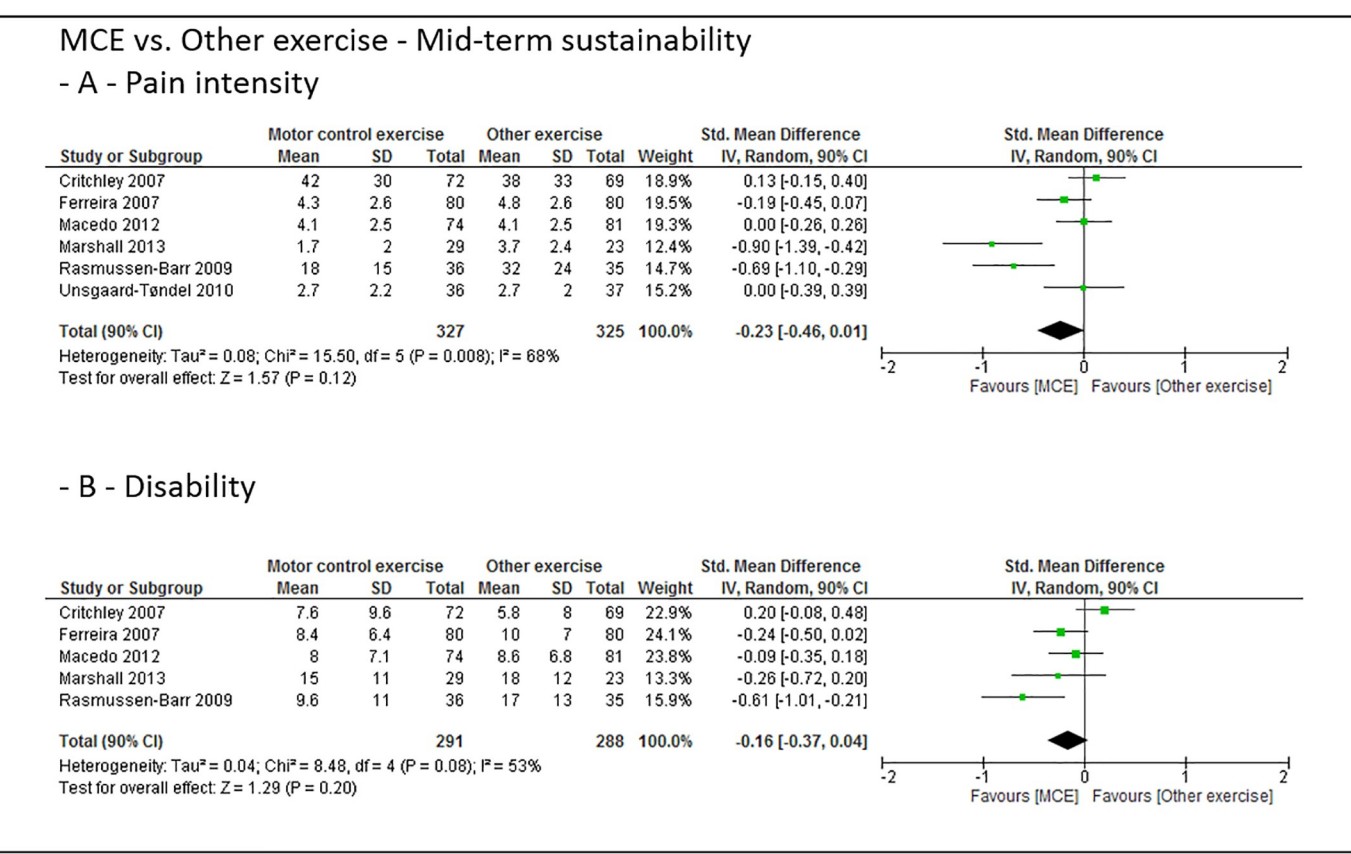

**Fig 6. Pooled effect sizes (standardized mean differences) for the outcomes pain intensity (-A-) and disability (-B-).** Analysis for the mid-term sustainability effects of motor control stabilisation exercise in comparison to other exercises. MCE, motor control stabilisation exercise; SD, standard deviation; CI, confidence interval.

lower quality studies have been demonstrated in other disease therapies, like depression [32]. More high-quality evidence is thus needed to prove our findings.

## Limitations at review level

The funnel plot analysis revealed an unclear but rather low risk of publication bias within our review. As the risk is nevertheless unclear, and the findings of the main analyses were heterogeneous, future study potentially affects the main findings towards positive effects of MCE compared to other interventions (most likely), no difference between MCE and other exercises (likely) or larger effect in other exercises (unlikely). We included studies in which the exercise intervention was completed. Although the scheduled intervention was definetely completed in each of the included studies, we do not know if the participants have continued with the exercises by their own. A certain uncertainty thus remains if the effects found are solely sustainability effects regarding the sustainability of the intervention effect or a mix of the sustainability effect and the sustainability of the intervention compliance.

The transfer of our results into practice may be limited against the proper definition of the studies' populations and therapy aims. Although all studies name long-term, follow-up or sustainability effects in chronic low back pain patients as the aim of the intervention, it remains unclear as to whether chronic, chronic-recurrent or even subacute participants were included.

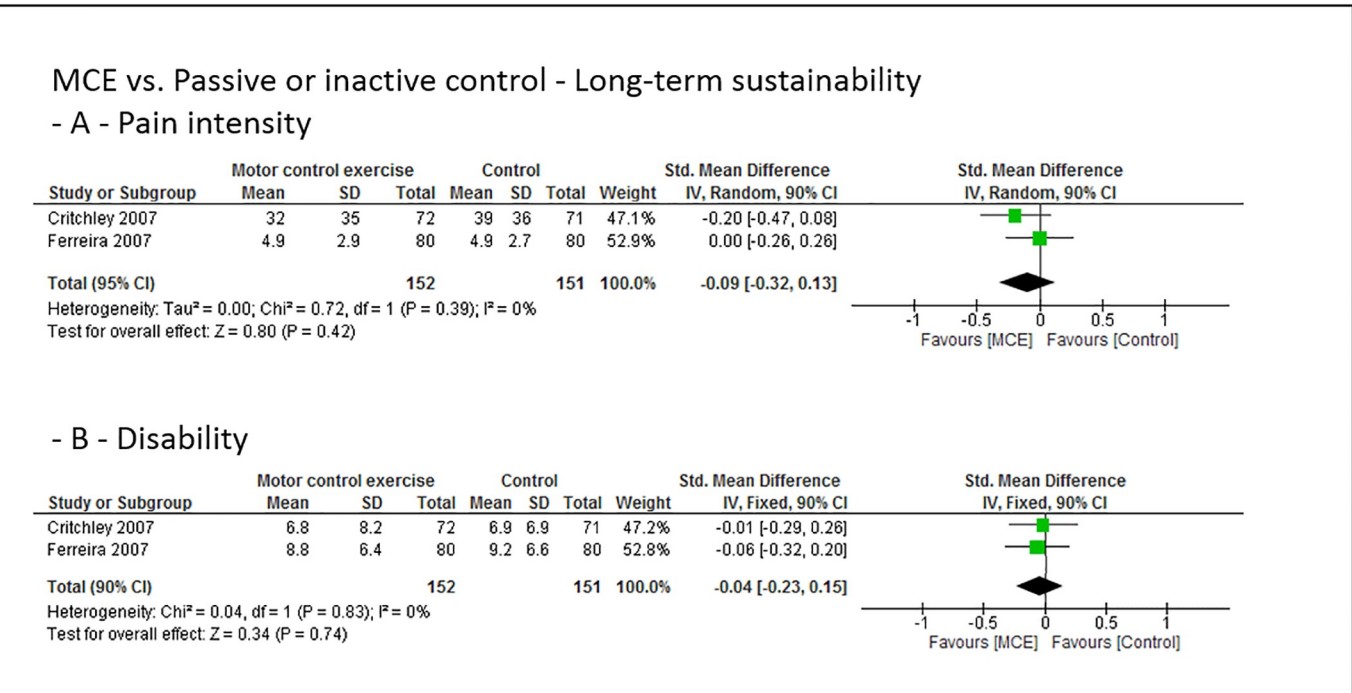

**Fig 7. Pooled effect sizes (standardized mean differences) for the outcomes pain intensity (-A-) and disability (-B-).** Analysis for the long-term sustainability effects of motor control stabilisation exercise in comparison to passive or inactive control. MCE, motor control stabilisation exercise; SD, standard deviation; CI, confidence interval.

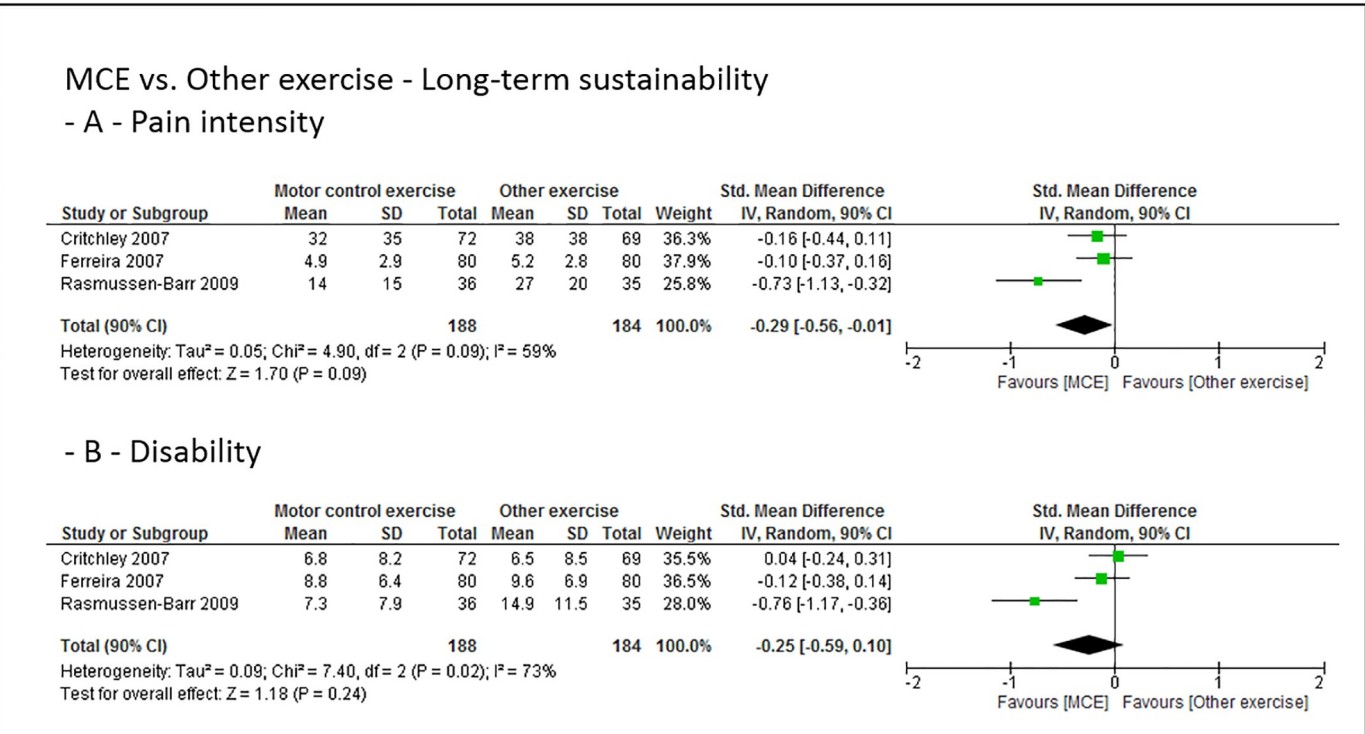

**Fig 8. Pooled effect sizes (standardized mean differences) for the outcomes pain intensity (-A-) and disability (-B-).** Analysis for the long-term sustainability effects of motor control stabilisation exercise in comparison to other exercises. MCE, motor control stabilisation exercise; SD, standard deviation; CI, confidence interval.

**Table 4. Individual studies' training characteristics.** All interventions and the respective comparators are described. MCE, motor control stabilisation exercise; N.A., not applicable.

| First author, year | Type MCE Intervention (MCE, CSE, Stabili, . . .) | Exercises (N): (Names) | Type comparator(s) | Training period (weeks) | Training Frequency (sessions per week) | Training duration (minutes per session) | Sets (number per exercise) | Repetitions (per set per exercise) | Rest (between sets per exercise; between exercises in seconds) |
|---|---|---|---|---|---|---|---|---|---|
| Bae, 2018 | CSE | 6: Abdominal drawing-in in 4-point kneeling and supine position, Opposite upper and lower extremity lift in quadruped position, Straight leg raise exercise in prone position, Supine lower extremity extender in supine position, Straight leg raise exercise in supine position, Horizontal side-support exercise in side lying position | Assisted sit-up exercise | 4 | 3 | 30 | N.A. | N.A. | N.A. |
| Critchley, 2007 | Spinal Stabil | N.A.: individual transversus abdominis and lumbar multifidus muscle training followed by group exercises that challenged spinal stability. Exercises were tailored to assessment findings and progressed within participants' ability to maintain a stable and minimally painful spine. The exercise program aimed to improve trunk muscle motor control | Physio, Pain Management | N.A. | 8 | 90 | N.A. | N.A. | N.A. |
| Ferreira 2007 | MCE | N.A.: Improving function of specific trunk muscles thought to control inter-segmental movement of the spine, including transversus abdominis, multifidus, the diaphragm and pelvic floor muscles (Richardson) | General exercise, Spinal manipulation therapy | 8 | 12 | N.A. | N.A. | N.A. | N.A. |
| Giesche 2017 | Sensorimotor Stabili in add to MMST | N.A.: Exercises in lying, sitting and standing positions | MMST | 2 | 7 | 60 | N.A. | N.A. | N.A. |
| Kofotolis, 2016 | Pilates | 16: Roll down, mermaid, spine stretching, pelvic curl, criss-cross, double leg stretch, hundreds, double knee folds, table top, swimming, swan, cat stretch, child's pose, hips stretch | General strengthening/ stabilisation exercise, control | 8 | 3 | 60 | 2 (until week 4), then 3 | 15 (week 1–2), 20 (w 3–4), 15 (5–6), 20 (7–8) | 2 |

*(Continued)*

**Table 4.** (Continued)

| First author, year | Type MCE Intervention (MCE, CSE, Stabili, . . .) | Exercises (N): (Names) | Type comparator(s) | Training period (weeks) | Training Frequency (sessions per week) | Training duration (minutes per session) | Sets (number per exercise) | Repetitions (per set per exercise) | Rest (between sets per exercise; between exercises in seconds) |
|---|---|---|---|---|---|---|---|---|---|
| Macedo, 2012 | MCE | N.A.: Varying interindividual | General Graded Activity | 8 | 2 (first 4 weeks), 1 (rest) | 60 | 1 | 10 | N.A. |
| Marshall, 2013 | MCE & Pilates | 8: Whole body stretching; Skilled abdominal contractions and postural training; Side lying trunk; Prone lying trunk; Hip-specific exercises; Upper and lower limb; Full body exercises; Whole body stretching | Stretching and cycling | 8 | 3 | 55 | N.A. | N.A. | N.A. |
| Rasmussen-Barr, 2003 | Stabil | 6–8: motor control, supine crooked-lying, four-point kneeling, prone, sitting and standing | Manual therapy | 6 | 1 supervised; 7 home-based | 45 supervised, 15 | 3 | 15 | N.A. |
| Rasmussen-Barr, Eva, 2009 | Graded Stabil | 6–8: N.A. | 30-minute walk every day | 8 | 1 supervised; 7 home-based | 45 supervised, 15 self-admin | 3 | 15 | N.A. |
| Unsgaard-Tondel, 2010 | Sling Training | N.A.: Sling training | Low-load MCE (feedback) and General exercise | 8 | 1 | 40 | N.A. | N.A. | N.A. |

In chronic-recurrent and subacute patients, the effects are rather sustainability of the therapy but effects of tertiary prevention. Only limited evidence is available if tertiary prevention is effective when adopting exercise in general, and MCE in particular[33,34]. To provide further evidence, running RCTs should differ in their reports between tertiary (recurrence) prevention, long-term effects and sustainability[35].

## Sensitivity of the interventions' name

The interventions of the studies included into our meta-analysis is called "motor control stabilisation exercise". Motor control exercises are classically defined as core-specific dynamic stabilisation exercises with an a priori education on deep trunk muscles activation and/or the control of deep muscles activation during exercising [36]. We only included studies with

**Table 5. Outcomes of the sensitivity meta-regressions.** For each single analysis, effect sizes, number of included effect sizes, homogeneity, the regression coefficient B, its confidence interval (CI) and the corresponding p-value are displayed.

| Model (independent variable) | Mean effect size | N effect sizes included | Homogeneity Q | B | 95% CI | p-value |
|---|---|---|---|---|---|---|
| Intervention: Duration [weeks] | 1.01 | 8 | 2.1 | -.09 | -.22, .03 | .15 |
| Intervention: Frequency [$N_{Trainings}$/weeks] | 1.00 | 8 | .0001 | .0007 | -.11, .11 | .99 |
| Intervention: Ratio sustainability:training | 1.2 | 15 | 1.3 | -.04 | -.11, .03 | .25 |
| Intervention: total dose [minutes] | 1.0 | 8 | .87 | -.0004 | -.001, .0004 | .35 |
| Study quality: Pedro [points] | 1.12 | 15 | 6.1 | -.24 | -.43, -.05 | .014 |

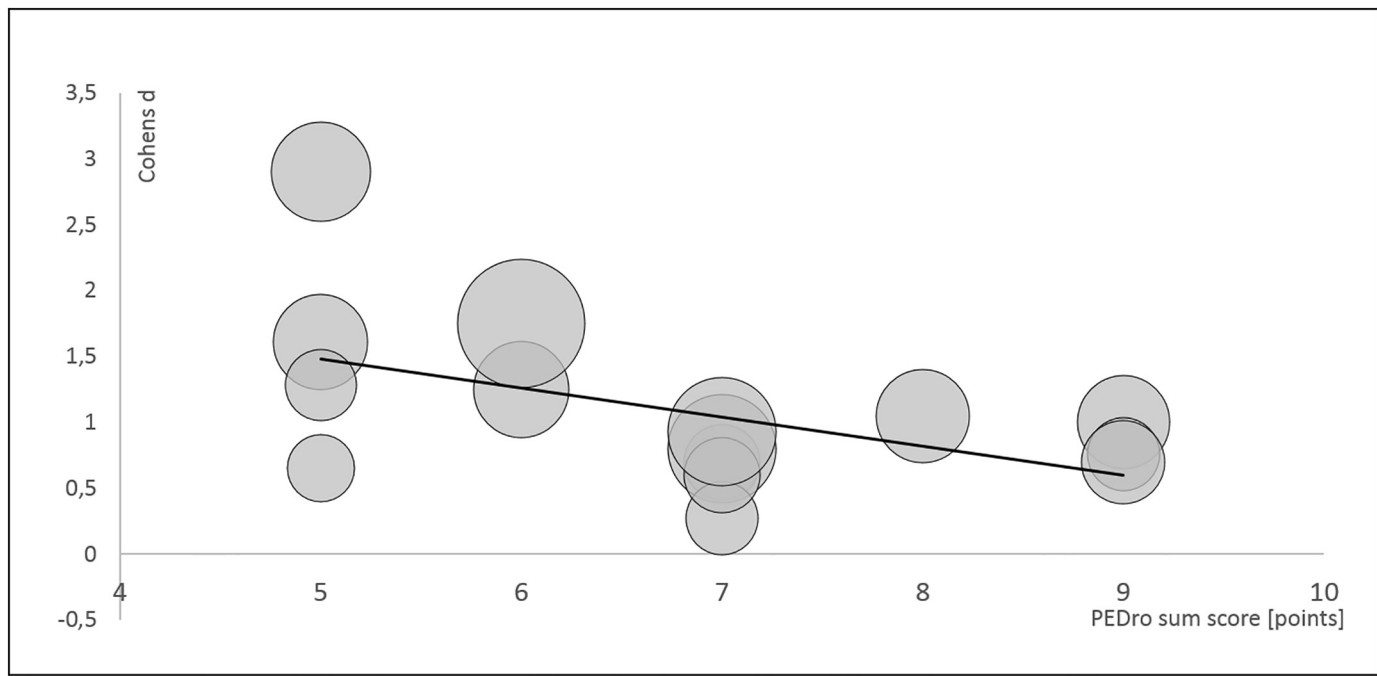

**Fig 9. Meta-regression bubble plot for the dependent variable Cohens d, independent variable PEDro sum score and weighting (illustrated by the size of the bubbles).**

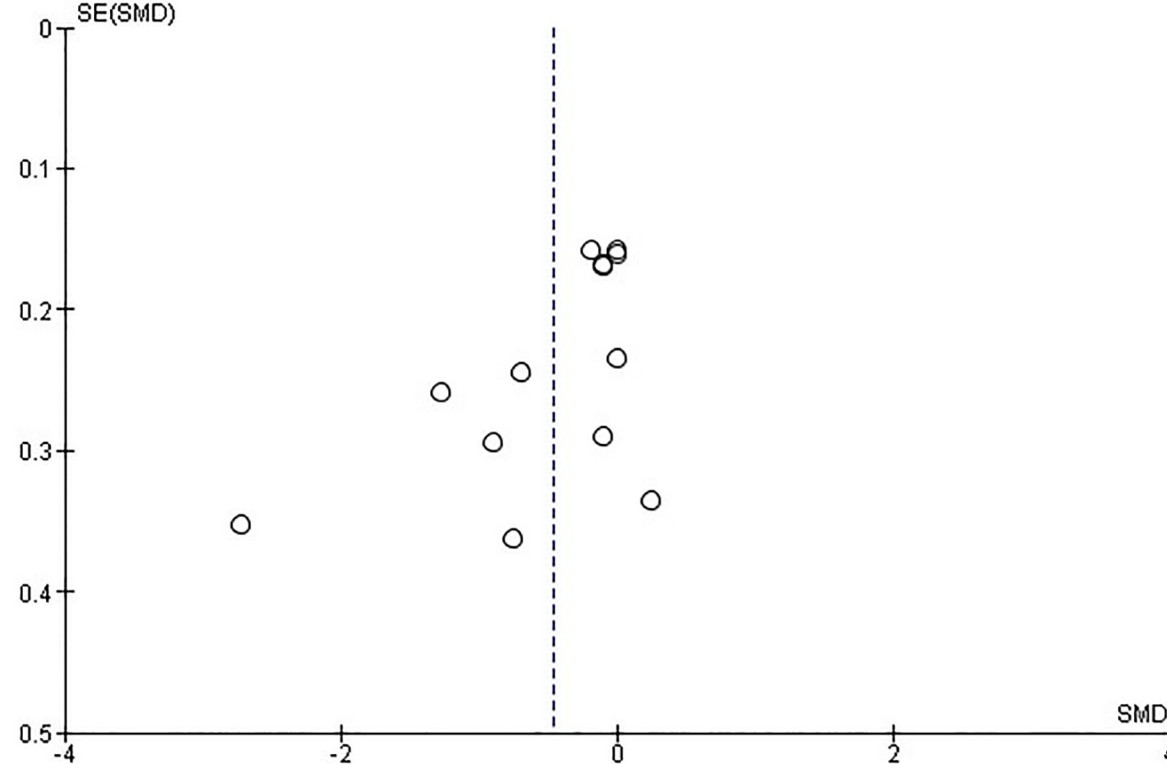

**Fig 10. Funnel plot of all studies included.** Each first sustainability SMD (standard mean differences and their belonging SE (standard errors) are plotted.

dynamic/exercise parts. When solely stabilisation exercises without pre-conditioning are performed, they are often called "coordination", "stabilisation" [4], "sensorimotor"[35] or even as well "motor control"[5] exercise. As described above, the term "motor control exercise"may be slightly too sensitive for the interventions included into our review. In contrary, the terms "sensorimotor", "coordination" and "stabilisation" training/exercise may be too general. Consequently, we name the intervention "motor control stabilisation exercise" to highlight that the stabilisation/active/dynamic parts of the originally described as "motor control exercise"-theorem are adopted. Nevertheless, the intervention could also be called "core-specific stabilisation" or "sensorimotor exercise".

## Perspective

We found low to moderate quality evidence for a sustainable positive effect of motor control stabilisation exercise on pain and disability in low back pain patients when compared to any control. The subgroups effects are less clear, and no clear direction of short vs. mid vs. long-term, nor of the type or dose of the comparator, is given. Low-quality studies overestimate the effects of motor control stabilisation exercises. Further high-quality studies are needed to prove or adopt our findings.

## Supporting information

**S1 Table. PRISMA checklist.**
(DOC)

## Author Contributions

**Conceptualization:** Daniel Niederer, Juliane Mueller.

**Data curation:** Daniel Niederer, Juliane Mueller.

**Formal analysis:** Daniel Niederer, Juliane Mueller.

**Investigation:** Daniel Niederer, Juliane Mueller.

**Methodology:** Daniel Niederer, Juliane Mueller.

**Project administration:** Daniel Niederer, Juliane Mueller.

**Validation:** Daniel Niederer.

**Visualization:** Daniel Niederer, Juliane Mueller.

**Writing – original draft:** Daniel Niederer.

**Writing – review & editing:** Daniel Niederer, Juliane Mueller.

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
