## [Decision Letter · Decision Letter 0]

12 Dec 2019

PONE-D-19-29553

Sustainability effects of motor control stabilisation exercises on pain and function in chronic nonspecific low back pain patients: A systematic review with meta-analysis and meta-regression

PLOS ONE

Dear Dr Niederer,

Thank you for submitting your manuscript to PLOS ONE. After careful consideration, we feel that it has merit but does not fully meet PLOS ONE’s publication criteria as it currently stands. Therefore, we invite you to submit a revised version of the manuscript that addresses the points raised during the review process.

Two peer reviewers have provided extensive review of your manuscript. I would appreciate if you could respond to each of their comments. They have both raised the issue about the definition of sustainability. Other methodological choices and further definitions are queried which require a response. In relation to the comment how this review is different to the Cochrane 2016 review, this review is clearly more up-to-date. I will leave it to you to highlight other differences / respond to the reviewer.

We would appreciate receiving your revised manuscript by 11th February 2019. To enhance the reproducibility of your results, we recommend that if applicable you deposit your laboratory protocols in protocols.io, where a protocol can be assigned its own identifier (DOI) such that it can be cited independently in the future. For instructions see: http://journals.plos.org/plosone/s/submission-guidelines#loc-laboratory-protocols

We look forward to receiving your revised manuscript.

Kind regards,

Dr. Leica S. Claydon-Mueller

Academic Editor

PLOS ONE

Journal Requirements:

2. Please amend either the title on the online submission form (via Edit Submission) or the title in the manuscript so that they are identical.

3. Please amend your manuscript to include your abstract after the title page.

Reviewers' comments:

Reviewer's Responses to Questions

**Comments to the Author**

1. Is the manuscript technically sound, and do the data support the conclusions?

Reviewer #1: No

Reviewer #2: Yes

2. Has the statistical analysis been performed appropriately and rigorously? 

Reviewer #1: No

Reviewer #2: Yes

3. Have the authors made all data underlying the findings in their manuscript fully available?

Reviewer #1: Yes

Reviewer #2: Yes

4. Is the manuscript presented in an intelligible fashion and written in standard English?

Reviewer #1: No

Reviewer #2: Yes

5. Review Comments to the Author

Reviewer #1: I thank the authors for the opportunity to review this manuscript. The authors set out to investigate whether motor control exercises lead to a sustainable improvements in pain intensity and disability in patients with chronic nonspecific low back pain, and to investigate to what extent the time after training cessation, study quality, and training characteristics modify these effects. I commend the authors for the amount of work that has gone into this review, however, there are several issues with the definitions and aims of this review that make it hard to follow and distinguish it from previous reviews of motor control for chronic LBP.

Major comments

Definitions and aims

I am confused about the term 'sustainability' throughout the paper. After reading the title and abstract I thought it was referring to how improvements in motor control are sustained over time. After reading the aim at the end of the introduction I thought it was referring to the long-term effects of motor control. In the methods, it appears to be neither as the authors include short-, medium- and long-term outcomes. With this in mind I am unsure how this review differs from the 2016 Cochrane review on this topic (which included 29 studies)- which also investigated the effects of motor control for chronic LBP at various time points

Results

-Including non-randomised trials does not align with recommendations from Cochrane. I suggest removing these two studies

Discussion

-Line 224-253: the authors should try to avoid repeating the results in the discussion. Instead, the discussion should summarise what the results mean.

Minor comments

Introduction

-Page 8, lines 32-47: there seems to be a lot of repetition here. I encourage the authors to condense this section by being more concise.

-Page 9, line 48: 'lack of uniform definitions' is ambiguous in this context. What definitions are the authors referring to?

-Page 9, line 56-60: this sentence is hard to follow. I recommend splitting it in two. Likewise for the sentence on lines 62-65

Methods

-Having key inclusion/exclusion criteria in the text would be helpful

-The authors used both the PEDro scale and Cochrane risk of bias tool to assess risk of bias. Usually reviews only include one.

Results

-Lines 211-222: could this section be condensed? There seems to be a lot of repetition

-Lines 223-228: This should come earlier where the characteristics of individual studies is outlined

Reviewer #2: I would like to thank the authors for a well written and presented manuscript which addresses an important clinical issue

Overall, it would be good if the authors could provide a clearer rationale for their definition of sustainability and whether the authors are actually looking at this phenomenon

in order for an exercise to be sustainable - patients must continue with the exercise and or perform the exercise but it is not clear whether any studies look at issues of compliance/adherence to the exercise either during the trial period or more importantly after the completion of the trial. This issue is also not discussed and should form part of the discussion around sustainability (if patients do not undertake exercise once a trial finishes then the potential for sustainable effects on pain are potentially lessened)

In the introduction and background - the authors should try to be consistent in reporting the outcomes and magnitude of the effect size and differences for studies reported (as well as timepoints)

It is also unclear how the authors suggested that sensorimotor training is one of the most effective forms of exercise - does the evidence support this when comparing different forms of exercise and in what populations and timepoints (important to highlight when the focus is on sustainability)

The authors also should try to provide a clear definition of the main constructs - both sensorimotor training and sustainability effects - a number of terms are used without reference to source. It perhaps needs to be clearer how the definitions used differ from others and or how exercise programmes that incorporate elements of sensorimotor and other forms of exercise were dealt with

The authors should provide a reference for the PRISMA in methods approach

in terms of the search strategy - it does not appear that specific types of interventional approaches (e.g. Pilates were used as search terms - can the authors please justify this in terms of not using specific terms for exercise approaches?)

In terms of the PICO - can the authors please clarify what is defined as active and passive treatment for LBP?

In the inclusion and results section - it appears that some (x2) studies have patients have less than 12 weeks duration (which is not defined as chronic?)

Looking at the main results and the forest plots - it is evident that one article Kofotolis (2016) has recorded a much larger effect across all measures than all other studies - it appears that this result may of had quite a significant effect on the overall findings and conclusions. This study was of relatively poor quality and the authors note that study quality (Pedro score) and effect estimate were inversely related - I think it would be worth a specific discussion around the findings and limitations of this paper esp. when looking at the potential effect on the overall results

it is also a study on the effects of Pilates - can the authors please comment on why they did not identify and include more studies on Pilates comparing exercises?

overall, the limitations are broadly discussed and relevance to research and practice identified.

6. PLOS authors have the option to publish the peer review history of their article (what does this mean?). If published, this will include your full peer review and any attached files.

Reviewer #1: Yes: Joshua Zadro

Reviewer #2: No

---

## [Author Response · Author response to Decision Letter 0]

16 Dec 2019

Dear editor, dear reviewer

Thank you for the valuable comments, which we all accounted for the revision of our manuscript to improve its quality Please find our point-to-point-responses to your queries below.

Reviewer #1: I thank the authors for the opportunity to review this manuscript. The authors set out to investigate whether motor control exercises lead to a sustainable improvements in pain intensity and disability in patients with chronic nonspecific low back pain, and to investigate to what extent the time after training cessation, study quality, and training characteristics modify these effects. I commend the authors for the amount of work that has gone into this review, however, there are several issues with the definitions and aims of this review that make it hard to follow and distinguish it from previous reviews of motor control for chronic LBP.

- Thanks for your input. By considering all your raised comments below, we hope to be more precise and clarify the difference to already existing reviews. Please see all comments below.

Major comments

Definitions and aims

1. I am confused about the term 'sustainability' throughout the paper. After reading the title and abstract I thought it was referring to how improvements in motor control are sustained over time. After reading the aim at the end of the introduction I thought it was referring to the long-term effects of motor control. In the methods, it appears to be neither as the authors include short-, medium- and long-term outcomes. With this in mind I am unsure how this review differs from the 2016 Cochrane review on this topic (which included 29 studies) - which also investigated the effects of motor control for chronic LBP at various time points

- This is an important point. Indeed, we differentiated between 1. long-term effects (where the exercise intervention was performed until the assessment) and 2. sustainability effects, where the exercise intervention was completed a certain time (at least four weeks in our case) before the assessment of the outcome.

The term “training cessation” we used in the manuscript, is not very precise and was replaced. We further tried to be more precise in the thorough manuscript to prevent further misunderstanding and to more clearly highlight the difference from our meta-analysis to the 2016 Cochrane review. In the latter, the classic long-term and sustainability effects were mixed.

The following changes were made in the manuscript:

We replaced “training cessation” by “exercise intervention completion” in the thorough manuscript.

- Abstract: “...with at least one pain intensity and disability outcome assessment at a follow-up (sustainability) time point of ≥ 4 weeks after exercise intervention completion. “line 37

- Introduction: Here, major changes were made in the lines 88+ 

- Methods: Table 1:

Follow-up length > 3 weeks after exercise intervention completion Continous exercise intervention until follow-up meassurement

Intervention stabilization exercises/training interventions with a defined completion time 

- Discussion: “Thus, a mix of sustainability effects and effects directly assessed during the intervention or directly after the intervention completion are mixed.” lines 368+

Although the scheduled intervention was completed, we do not know if the participants do continue with the exercises or not. We thus added to the limitations (review level): “We included studies in which the exercise intervention was completed. Although the sheduled intervention was definitely completed in each of the included studies, we do not know if the participants have continued with the exercises by their own. A certain uncertainty thus remains if the effects found are solely sustainability effects regarding the sustainability of the intervention effect or a mix of the sustainability effect and the sustainability of the intervention compliance.” lines 417+

 Please also refer to the answers on comment 1 from reviewer 2.

2. Results - Including non-randomised trials does not align with recommendations from Cochrane. I suggest removing these two studies

- The authors discussed this topic as well prior to data analysis. As we performed detailed sensitivity analyses on (independent variable) the impact of risk of bias and study quality, we decided to include as well non-randomized trials. Following your hint, we added this topic to the limitations: “This finding is most likely attributed to the lack of adopting a randomized design (2 studies) as well as to the lack of participant and study personnel blinding.” lines 406+

3. Discussion - Line 224-253: the authors should try to avoid repeating the results in the discussion. Instead, the discussion should summarise what the results mean.

- After reading this section again, we must agree (embarrassingly). The section was re-written:

“We found that motor control stabilisation exercises lead, with low to moderate quality evidence, to a sustainable improvement in pain intensity and disability in chronic non-specific low back pain patients compared to an inactive or passive control group or compared to other exercises. Subgroup sensitivity analyses revealed less clear findings: some of the pooled effects reached significance, some not.” lines 342+

Minor comments

Introduction

4. Page 8, lines 32-47: there seems to be a lot of repetition here. I encourage the authors to condense this section by being more concise.

- These sections were shortened accordingly. “Motor-control exercises [5] and Pilates-based stabilization exercises [6] have been shown to be superior to minimal intervention and provide at least similar outcomes to other forms of exercises [5,6]. Core-stability exercises [7] and back pain-oriented stabilization exercises [8] are more effective than general exercises. In general, strength/resistance and coordination/stabilisation exercise programs seem to be superior to other interventions in the treatment of chronic low back pain [4]. Taken together, and proofed in a recent network meta-analysis on the direct comparison of exercise types [9] sensorimotor training is – regarding the outcome pain - one of the most, and -regarding physical function – the most effective active regimens for chronic low back pain treatment”. lines 55+

5. Page 9, line 48: 'lack of uniform definitions' is ambiguous in this context. What definitions are the authors referring to?

- changed to “The various exercises summarized under “sensorimotor/stability/motor control” lines 69+

6. Page 9, line 56-60: this sentence is hard to follow. I recommend splitting it in two. Likewise for the sentence on lines 62-65

- As recommended, splitted into two sentences: “Classically, motor control exercises contain a pre-education on deep trunk muscles activation and/or the control of deep muscles activation during exercising. in contrast, different definitions and/or definitions with overlaps to non-dynamic motor control situations are often summarized under the term motor control, the pooled effects of (not only but also) long-term effects may have been over- or underestimated.”

And

“In contrast, different definitions and/or definitions with overlaps to non-dynamic motor control situations are often summarized under the term motor control, the pooled effects of (not only but also) long-term effects may have been over- or underestimated.” 

Methods

7. Having key inclusion/exclusion criteria in the text would be helpful

- To prevent redundancy, we added the table with the corresponding information directly under the corresponding section.

8. The authors used both the PEDro scale and Cochrane risk of bias tool to assess risk of bias. Usually reviews only include one.

- The Cochrane RoB tool was used to assess the risk of Bias, the Pedro scale to assess the overall study quality. Here, we followed the recommendations in the Cochrane handbook (chapter 7.1.2): The RoB should not be derived by the PEDro scale. 

- Further: “The lack of a theoretical framework underlying the concept of ‘quality’ assessed by these scales resulted in tools mixing different concepts such as risk of bias, imprecision, relevance, applicability, ethics, and completeness of reporting.” (from chapter 7.1.2). As the study quality itself was one of the predictors in our sensitivity analysis, we decided to assess bot the RoB and quality separately.

9. Results -Lines 211-222: could this section be condensed? There seems to be a lot of repetition

- As recommended, this section is condensed. 

10. Lines 223-228: This should come earlier where the characteristics of individual studies is outlined

- As the training specifics/characteristics are (usually) not reported in meta-analysis (at least not so detailed) and we used them (in particular) for the sensitivity meta-regressions, we added the belonging information here.

Reviewer #2: I would like to thank the authors for a well written and presented manuscript which addresses an important clinical issue

1. Overall, it would be good if the authors could provide a clearer rationale for their definition of sustainability and whether the authors are actually looking at this phenomenon

in order for an exercise to be sustainable - patients must continue with the exercise and or perform the exercise but it is not clear whether any studies look at issues of compliance/adherence to the exercise either during the trial period or more importantly after the completion of the trial. This issue is also not discussed and should form part of the discussion around sustainability (if patients do not undertake exercise once a trial finishes then the potential for sustainable effects on pain are potentially lessened)

- This is an important point. Indeed, we differentiated between 1. long-term effects (where the exercise intervention was performed until the assessment) and 2. sustainability effects, where the exercise intervention was completed a certain time (at least four weeks in our case) before the assessment of the outcome. The term “training cessation” we used in the manuscript, is not very precise and was replaced. We further tried to be more precise in the thorough manuscript to prevent further misunderstanding and to more clearly highlight the difference from our meta-analysis to the 2016 Cochrane review. In the latter, the classic long-term and sustainability effects were mixed.

- We replaced “training cessation” by “exercise intervention completion” in the thorough manuscript.

- In the Abstract: “...with at least one pain intensity and disability outcome assessment at a follow-up (sustainability) time point of ≥ 4 weeks after exercise intervention completion. “ line 37

- Introduction: Here, major changes were made in the lines 88+

- Methods: Table 1 was changed

- Discussion: “Thus, a mix of sustainability effects and effects directly assessed during the intervention or directly after the intervention completion are mixed.” lines 368+

- Although the sheduled intervention was completed, we do not know if the participants do continue with the exercises or not. We thus added to the limitations (review level): “We included studies in which the exercise intervention was completed. Although the sheduled in-tervention was definitely completed in each of the included studies, we do not know if the participants have continued with the exercises by their own. A certain uncertainty thus remains if the effects found are solely sustainability effects regarding the sustainability of the intervention effect or a mix of the sustainability effect and the sustainability of the intervention compliance.” lines 417+

- Please also refer to the answers on comment 1 from reviewer 1.

2. In the introduction and background - the authors should try to be consistent in reporting the outcomes and magnitude of the effect size and differences for studies reported (as well as timepoints)

3. It is also unclear how the authors suggested that sensorimotor training is one of the most effective forms of exercise - does the evidence support this when comparing different forms of exercise and in what populations and timepoints (important to highlight when the focus is on sustainability)

- These information was precised (a great network meta-analysis was recently published in this topic, please refer to the reference (new) number 9). The section was re-written accordingly: “Motor-control exercises [5] and Pilates-based stabilization exercises [6] have been shown to be superior to minimal intervention and provide at least similar outcomes to other forms of exercises [5,6]. Core-stability exercises [7] and back pain-oriented stabilization exercises [8] are more effective than general exercises. In general, strength/resistance and coordination/stabilisation exercise programs seem to be superior to other interventions in the treatment of chronic low back pain [4]. Taken together, and proofed in a recent network meta-analysis on the direct comparison of exercise types [9] sensorimotor training is – regarding the outcome pain - one of the most, and -regarding physical function – the most effective active regimens for chronic low back pain treatment” lines 55+

4. The authors also should try to provide a clear definition of the main constructs - both sensorimotor training and sustainability effects - a number of terms are used without reference to source. It perhaps needs to be clearer how the definitions used differ from others and or how exercise programmes that incorporate elements of sensorimotor and other forms of exercise were dealt with

- Sustainability effects: please refer to the changes and comments on query number 1.

- sensorimotor training: This is an important topic. We added a completely new chapter on this information into the discussion:

“Sensitivity of the intervention name

The interventions included into our meta-analysis is called “motor control stabilisation exercise”. Motor control exercises are classically defined as core-specific dynamic stabilisation exercises with an a priori education on deep trunk muscles activation and/or the control of deep muscles activation during exercising [24]. We only included studies with dynamic/exercise parts. When solely stabilisation exercises without pre-conditioning are performed, they are often called “coordination”, “stabilisation” [29] , „sensorimotor“ [23] or even as well „motor control“ [28] exercise. As described above, the term „motor control exercise“ may be slightly too sensitive for the interventions included into our review. In contrary, the terms “sensorimotor”, “coordination” and “stabilisation” training/exercise may be too general. Consequently, we name the intervention “motor control stabilisation exercise” to highlight that the stabilisation/active/dynamic parts of the originally described as “motor control exercise”-theorem are adopted. Nevertheless, the intervention could also be called “core-specific stabilisation” or “sensorimotor exercise”. lines 431+

5. The authors should provide a reference for the PRISMA in methods approach

in terms of the search strategy - it does not appear that specific types of interventional approaches (e.g. Pilates were used as search terms - can the authors please justify this in terms of not using specific terms for exercise approaches?)

- The reference was provided

- Exactly, we have not searched for specific “names”. We explicitely searched for the underlying mechanism of coordination/sensorimotor/balance training. Thus, the training specifis were: stabili* OR sensorimotor OR “motor control” OR neuromuscular OR perturbation. That includes Pilates trainings with a particular stabilisation focus.

6. In terms of the PICO - can the authors please clarify what is defined as active and passive treatment for LBP?

- We added a definition: “(no active involvement of the patient, mostly massage therapy, manual therapy, and thermotherapy) lines 98-99

7. In the inclusion and results section - it appears that some (x2) studies have patients have less than 12 weeks duration (which is not defined as chronic?)

- We agree. One study had > 6 and the second > 8 weeks as inclusion criteria. Some of the participants were thus “only” sub-acute and not chronic. After study completion, all participants were chronic as a certain time has passed since inclusion. However, we changed the inclusion criterion “duration” to “non-acute (sub-acute or chronic > 6 weeks of duration at the time of study inclusion)” to be more precise. Please refer to table 1.

8. Looking at the main results and the forest plots - it is evident that one article Kofotolis (2016) has recorded a much larger effect across all measures than all other studies - it appears that this result may of had quite a significant effect on the overall findings and conclusions. This study was of relatively poor quality and the authors note that study quality (Pedro score) and effect estimate were inversely related - I think it would be worth a specific discussion around the findings and limitations of this paper esp. when looking at the potential effect on the overall results

- We added this limitation into the corresponding section: “This finding is most likely attributed to the lack of adopting a randomized design (2 studies) as well as to the lack of participant and study personnel blinding or to the fact that most of our significant findings were attributed to only two studies with large effects [26,27]“ lines 406-408

9. it is also a study on the effects of Pilates - can the authors please comment on why they did not identify and include more studies on Pilates comparing exercises?

- Pleased refer to the answer in the comments on your query number 5

10. overall, the limitations are broadly discussed and relevance to research and practice identified.

Thank you.

---

## [Editor Report · Decision Letter 1]

19 Dec 2019

Sustainability effects of motor control stabilisation exercises on pain and function in chronic nonspecific low back pain patients: A systematic review with meta-analysis and meta-regression

PONE-D-19-29553R1

Dear Dr. Niederer,

We are pleased to inform you that your manuscript has been judged scientifically suitable for publication and will be formally accepted for publication once it complies with all outstanding technical requirements.

With kind regards,

Leica S. Claydon-Mueller

Academic Editor

PLOS ONE
---

## [Editor Report · Acceptance letter]

31 Dec 2019

PONE-D-19-29553R1 

Sustainability effects of motor control stabilisation exercises on pain and function in chronic nonspecific low back pain patients: A systematic review with meta-analysis and meta-regression 

Dear Dr. Niederer:

I am pleased to inform you that your manuscript has been deemed suitable for publication in PLOS ONE. Congratulations! Your manuscript is now with our production department. 

With kind regards,

on behalf of

Dr. Leica S. Claydon-Mueller 

Academic Editor

PLOS ONE